# A Candidate DNA Vaccine Encoding the Native SARS-CoV-2 Spike Protein Induces Anti-Subdomain 1 Antibodies

**DOI:** 10.3390/vaccines11091451

**Published:** 2023-09-03

**Authors:** Anders Frische, Vithiagaran Gunalan, Karen Angeliki Krogfelt, Anders Fomsgaard, Ria Lassaunière

**Affiliations:** 1Department of Virus & Microbiological Special Diagnostics, Statens Serum Institut, 2300 Copenhagen, Denmark; afi@ssi.dk (A.F.); vigu@ssi.dk (V.G.); kak@ssi.dk (K.A.K.); afo@ssi.dk (A.F.); 2Section of Molecular and Medicinal Biology, Department of Science and Environment, Roskilde University, 4000 Roskilde, Denmark; 3Infectious Diseases Unit, Clinical Institute, University of Southern Denmark, 5230 Odense, Denmark

**Keywords:** SARS-CoV-2, DNA vaccine, epitope mapping, microarray, neutralizing antibodies

## Abstract

The ideal vaccine against viral infections should elicit antibody responses that protect against divergent strains. Designing broadly protective vaccines against SARS-CoV-2 and other divergent viruses requires insight into the specific targets of cross-protective antibodies on the viral surface protein(s). However, unlike therapeutic monoclonal antibodies, the B-cell epitopes of vaccine-induced polyclonal antibody responses remain poorly defined. Here we show that, through the combination of neutralizing antibody functional responses with B-cell epitope mapping, it is possible to identify unique antibody targets associated with neutralization breadth. The polyclonal antibody profiles of SARS-CoV-2 index-strain-vaccinated rabbits that demonstrated a low, intermediate, or high neutralization efficiency of different SARS-CoV-2 variants of concern (VOCs) were distinctly different. Animals with an intermediate and high cross-neutralization of VOCs targeted fewer antigenic sites on the spike protein and targeted one particular epitope, subdomain 1 (SD1), situated outside the receptor binding domain (RBD). Our results indicate that a targeted functional antibody response and an additional focus on non-RBD epitopes could be effective for broad protection against different SARS-CoV-2 variants. We anticipate that the approach taken in this study can be applied to other viral vaccines for identifying future epitopes that confer cross-neutralizing antibody responses, and that our findings will inform a rational vaccine design for SARS-CoV-2.

## 1. Introduction

On 12 December 2019, severe acute respiratory syndrome virus 2 (SARS-CoV-2) emerged, presumably through a zoonotic spillover from an animal in Wuhan, China [1]. The early release of the viral whole genome sequence and the characterization of the viral entry mediated by the spike protein facilitated critical research within SARS-CoV-2 spike/ACE2 interactions, pathogenesis and possible intervention strategies [2]. The spike protein occurs on the viral surface as homotrimers comprising the non-covalently associated subunits S1 and S2. The receptor binding domain (RBD), positioned at the S1 subunit, can, through a hinge-like structure, attend either a ‘down’, conformational masked mode for immune evasion, or an ‘up’ receptor-accessible conformation [3,4,5,6,7]. To enter a cell, the spike protein must bind to the host receptor ACE2 through the RBD, followed by cleavage at the S2’ cleavage site by cell surface proteases TMPRSS2, furin, or cathepsin (endosomal pathway) to release the fusion peptide [3,4,5,7,8,9,10,11,12]. The prefusion trimer is then destabilized and the S1 subunit is disconnected. This positions the fusion peptide on the S2 subunit to be inserted as a wedge into the cell membrane, followed by the formation of a six-helix bundle through heptad repeats 1 and 2, which brings the viral envelope and the host cell membrane into proximity, resulting in membrane fusion [3,5,13]. Considering the functional significance of the RBD, the S2’ cleavage site, and fusion elements, these spike protein domains are important targets for vaccine-induced antibodies to block/neutralize virus entry into the host cell.

Neutralizing antibodies may block viral entry into the host cell through a direct binding to the receptor binding motif, the steric hindrance of receptor binding, the locking of the RBD in a ‘down’ position, a spike-conformational change disturbance, or membrane fusion interference [6,14,15]. The RBD is a known target for infection-induced and vaccine-induced neutralizing antibodies. However, the RBD of SARS-CoV-2 maintains a ‘down’ mode at a higher frequency compared to SARS-CoV [3], leaving it less accessible for neutralizing antibodies. In addition, S1, and in particular the RBD, are the primary regions for mutations in the spike, leaving effective RBD neutralizing antibodies as being more strain-specific. For these reasons, the more preserved regions of the S1 and the highly conserved S2 domain (91% similarity to SARS-CoV [13]) could potentially be more prone to broadly neutralizing antibodies [13]. Studies have reported broadly neutralizing B-cell epitopes within the conformational active regions of the S1, such as the SD1 and the N-terminal domain [16,17,18,19,20,21,22,23]. In the S2 subunit, essential fusion elements such as the fusion peptide and the heptad repeats are also well-known targets for neutralizing antibodies and therapeutic drugs [5,13,24]. Defining the exact binding epitope of strong neutralizing antibodies is a key to rational vaccine design and antibody drug development.

Epitope mapping is a powerful approach that provides valuable insights into antibody binding patterns. Peptide microarrays can identify antigenic sites and immunogenic hotspots on proteins such as the SARS-CoV-2 spike protein. B-cells or secreted antibodies recognize these antigenic determinants, hereafter referred to as B-cell epitopes. Characterizing B-cell epitopes of monoclonal antibodies, and polyclonal responses in animal models and clinical cohorts, can be valuable knowledge for the rational vaccine design to obtain vaccines that induce broadly neutralizing antibodies [25,26,27], define the binding of therapeutic monoclonal antibodies to viral variants [28], and identify antibodies generating undesired clinical outcomes such as antibody-dependent enhancement [29,30]. It can also improve specificity in serological assays [27,29,31,32]. Immunization with peptides, binding strongly neutralizing antibodies, is believed to elicit equally strongly neutralizing antibodies [30]. Defining the B-cell epitopes of broadly protective anti-SARS-CoV-2 antibodies is essential for the next generation of intervention strategies.

During the SARS-CoV-2 pandemic, we developed a DNA vaccine that encodes the native spike protein of the SARS-CoV-2 index strain [33]. This plasmid DNA vaccine induced anti-spike binding IgG and neutralizing antibodies in mice, rabbits, and rhesus macaques, as well as inducing robust Th1-dominant cellular responses in small animals [33]. In the present study, we aim to elucidate the underlying antibody targets associated with broad cross-neutralization observed in a proportion of intramuscular, needle-free-vaccinated rabbits during the preclinical evaluation of this vaccine. Here, we used peptide microarray technology, with circular constrained peptides, to define immunogenic hotspots and identify B-cell epitopes associated with the ability to cross-neutralize different variants, often referred to as neutralization breadth.

## 2. Materials and Methods

### 2.1. DNA Vaccination and Study Population

We developed and performed a preclinical evaluation of a candidate DNA vaccine in different animal models, including rabbits [33]. Five ten-week-old New Zealand white rabbits each received 125 µg of a nanoplasmid vector three times at two-week intervals. This DNA vaccine encodes the full, unmodified SARS-CoV-2 index strain spike protein derived from the Wuhan-Hu-1 strain (MN908947). GeneArt (Thermo Fisher Scientific, Dreieich, Germany) synthesized the human codon optimized SARS-CoV-2 spike sequence with EcoRI and XhoI restriction enzyme sites for cloning into the NTC8685-eRNA41H vector backbone (Nature Technology Corporation, Lincoln, NE, USA). The Nature Technology Corporation produced the pNTC-Spike plasmid using their antibiotic-free RNA-OUT selection procedure in NTC4862 *E. coli* cells (DH5α attλ::P5/6 6/6-RNA-IN-SacV, Cmr). Vaccine stocks were supplied at 10 mg/mL in phosphate buffered saline (PBS).

The rabbits were immunized intramuscularly using a PharmaJet^®^ Stratis IM device. Serum samples taken 2 weeks after the last vaccination were included for analysis in the present study. No control animals were used in this study. The rabbits were kept in a temperature-regulated, pathogen-free environment at the Statens Serum Institut in Copenhagen, Denmark. The animals had continuous access to water and regular pellet food. Their cages included bedding and elements for environmental stimulation. All care and processes involving the animals adhered to Danish laws, which are aligned with the EU Directive 2010/63/EU 107 concerning the safeguarding of animals utilized in scientific research. The experiments were supervised by the laboratory animal veterinarians at Statens Serum Institut. The experiment received approval by The Animal Experimentation Council, the National Competent Authority within this field (approval number 2017-15-0201-01322; approval date 29 September 2017) and the Internal Animal Welfare committee at the animal facility at Statens Serum Institute (approval date 10 December 2021).

### 2.2. Microneutralization

The SARS-CoV-2 live virus microneutralization assay was performed as described using different SAS-CoV-2 variants [34,35]. Heat-inactivated serum samples were 2-fold diluted and incubated with 300 × TCID_50_ for 1 h at 37 °C, 5% CO_2_. Tissue culture plates that were seeded with a Vero E6 cell monolayer (10,000 cells/well) the preceding day were washed twice with 100 µL of Dulbecco’s phosphate buffered saline (DPBS) and overlaid with 100 µL of the serum–virus mixture. The inoculated cells were incubated at 37 °C, 5% CO_2_, for 24 to 32 h, depending on the SARS-CoV-2 variant. Quadruplicate wells with cells containing either 100 μL of 300 × TCID_50_ SARS-CoV-2 virus (no serum) or 100 μL of the virus diluent only were included on each microneutralization plate as the virus control and cell control, respectively. Microneutralization plates were washed twice with 100 µL of DPBS, and the SARS-CoV-2-infected monolayers were subsequently fixed with cold 80% (*v*/*v*) acetone in DPBS for 10 min. The replicating virus was quantified in an ELISA, targeting the nucleocapsid protein of SARS-CoV-2 using primary mouse anti-SARS-CoV-2 nucleocapsid protein mAb clone 7E1B, diluted at 1:4000 (cat. #BSM-41414M, Bioss Inc, Woburn, MA, USA) and a secondary antibody, a goat anti-mouse IgG peroxidase-conjugated polyclonal antibody, diluted at 1:10,000 (cat. #A16078, Invitrogen, Thermo Fisher, Waltham, MA, USA). The incubations with the primary and secondary antibodies were carried out for five minutes on an orbital shaker (300 rpm) at room temperature, followed by a one-hour incubation at 37 °C. The plates were washed three times with wash buffer (1% (*v*/*v*) Triton-X100, 0.2% Tween20 in DPBS) for 30 s before the primary and secondary antibody incubations. After the secondary antibody incubation, the plates were washed 5 times with the wash buffer and 3 times with deionized water (no soak) to remove all washing buffer debris. Then, 100 μL of TMB ONE substrate (Kem-en-tec, cat. #4380) was added to each well and the plates were incubated in the dark for 10 min at room temperature. The reaction was stopped with 100 μL of 0.2 M H_2_SO_4_ and the absorbance was read at 450 nm, with 620 nm as a reference, on a FluoStar Omega plate reader (BMG Labtech, Offenburg, Germany).

### 2.3. Microarray

The custom cyclic constrained peptide microarray, spanning the full-length spike protein of the SARS-CoV-2 index strain, was designed and manufactured by PEPperPRINT (Heidelberg, Germany). The serum antibody binding to the peptides on the microarray was performed according to the manufacturer’s instructions. In brief, the microarray was equilibrated with a washing buffer (DPBS with 0.05% Tween20, pH 7.4) for 15 min. The subarrays were subsequently blocked with a blocking buffer (Rockland Blocking Buffer MB-070, Rockland Immunochemicals, Pottstown, PA, USA) for 30 min. To assess unspecific reactions, the peptide array was pre-stained with a biotinylated secondary antibody and fluorophore-conjugated streptavidin, as described below, and scanned. The microarray was equilibrated for 15 min in a staining buffer (DPBS with 0.005% Tween20 and 10% blocking buffer, pH 7.4). The serum, diluted in the staining buffer, was added into the individual subarrays and incubated overnight at 4 °C. The following day, each subarray was washed twice with the wash buffer for 10 s, and a biotin-conjugated goat anti-rabbit IgG F(c) secondary antibody (cat. #A16122, Invitrogen, Thermo Fisher), diluted at 1:500 in the staining buffer, was added to each subarray and incubated for 45 min. The subarrays were washed twice as before and incubated for 45 min with streptavidin Alexa Fluor 647 conjugate (Thermo Fisher Scientific, USA), diluted at 1:750. The subarrays were washed as before, with the microarray slide dipped twice in a dipping buffer (1 mM Tris buffer, pH 7.4), dried using a centrifuge at 1000 rpm for 1 min, and scanned on an Agilent SureScan microarray scanner. The subarrays were equilibrated with the staining buffer for 15 min and subsequently stained with a PEPperCHIP^®^ anti-HA control antibody, diluted at 1:2000 in the staining buffer, and incubated for 45 min. The subarrays were washed, dipped twice in the dipping buffer, dried via centrifugation and scanned as described above. All volumes added to the subarrays were 400 µL, as appropriate for a microarray glass slide with four subarrays. All incubation steps were carried out in the provided cassette from PEPperPRINT, ensuring dark conditions, on an orbital shaker at 140 rpm. With the exception of the overnight incubation of the serum at 4 °C, all other incubations were at room temperature. The microarrays were analyzed using the MAPIX Analyzer Software v. 9.1.0.

### 2.4. Spike and RBD Indirect ELISA

White Nunc™ MaxiSorp™ plates were coated with 100 μL of 1 μg/mL recombinant SARS-CoV-2 spike ectodomain or RBD proteins, and incubated overnight at 4 °C. The recombinant proteins included SARS-CoV-2 D614G Spike ectodomain (Cat. #10587-CV)*,* SARS-CoV-2 B.1.351 Spike ectodomain (Cat. #10777-CV)*,* SARS-CoV-2 Spike RBD (Cat. #10500-CV), or SARS-CoV-2 B.1.351 Spike RBD (Cat. #10735-CV). All proteins were sourced from R&D Systems. The following day, the coated plates were emptied and, without washing, 150 μL of blocking buffer (Dilution Buffer pH 7.2, Cat #1322, SSI Diagnostica (Hillerød, Denmark) + 2% bovine serum albumin, and 0.1% Tween20) was added. The plates were subsequently incubated for 5 min on an orbital shaker (300 rpm) at room temperature, and then for 1 h at 37 °C. The plates were washed three times in a washing buffer (PBS with 0.05% Tween20), allowing for a 30-s soak each time. Then, 100 μL of serially diluted rabbit serum, diluted from 1:20 to 1:81,920 in the dilution buffer, were added, and the plates were incubated as described above. The plates were washed three times in the washing buffer with a 30 s soak, and 100 μL of a 1:2000 dilution of horse radish peroxidase-conjugated mouse-anti-rabbit IgG antibody (Cat. #A1949 Sigma-Aldrich, Darmstadt, Germany) was added. The plates were incubated as described above, followed by being washed five times with the wash buffer and three times with deionized water, with no soak. Lastly, the plates were treated with 100 μL of BM Chemiluminescent Substrate (Cat. # 11582950001, Sigma-Aldrich, Germany) and incubated for 10 min in a dark environment at room temperature. The resulting luminescence was recorded using a FLUOstar Microplate Reader (BMG LABTECH, Germany).

### 2.5. Total Antibody RBD ELISA

The Wantai SARS-CoV-2 Ab ELISA was performed according to the manufacturers’ instructions. The Wantai ELISA measures the total antibodies of all classes in a double antigen sandwich format. Firstly, 100 µL of the serially diluted rabbit sera (1:20 to 1:81,920) were added to wells that were pre-coated with the recombinant SARS-CoV-2 RBD antigen, and incubated for 30 min at 37 °C. The wells were washed five times and the horse radish peroxidase-conjugated SARS-CoV-2 antigen was added and incubated for 30 min at 37 °C. The wells were washed five times, followed by the addition of a chromogen solution, and incubated for 15 min at 37 °C. The reaction was stopped and the OD values were read on an ELISA reader.

### 2.6. Statistics and Calculations

#### 2.6.1. Microneutralization

Neutralization titers were calculated as the interception of a 4-parameter logistic curve fit of the OD values obtained from the 2-fold titration of the samples, starting from a 1:10 dilution to 1:1280, with a cut off. The 50% virus infection cut off was calculated from the virus control wells and the cell control wells as follows:$$50\%\;virus\;infection\;cutoff=\frac{\left(average\;OD\;of\;virus\;control\;wells\right)+(average\;OD\;of\;cell\;control\;wells)}{2}$$

Curve fitting was carried out using GraphPad Prism and the cut off calculations and titer calculations were carried out using Excel. The final titers are based on triplicate determinations.

#### 2.6.2. Microarray

The individual determinations were the fluorescence intensity at 635 nm minus the background at 635 nm. The area for background determination was manually set in Mapix when placing the grid. The fluorescence calculations were calculated as the mean of Δ-fluorescence from two identical peptides. The calculations were normalized according to the mean of a positive HA control (*n* = 48) included on each subarray. Active epitopes were defined as the fluorescence intensities above the cut off. The cut off was defined as
$$Cutoff=\bar{X}+SD\times f$$
where $\bar{X}$ is the mean of n negative control determinations (*n* = 96) on each subarray, *SD* is the standard deviation, and *f* is the standard deviation multiplier provided by Frey et al. [36]. The confidence interval was set to 99.0%. The negative control was 96 negative spots included on each subarray.

#### 2.6.3. Spike ELISA and RBD ELISA

The endpoint titers were calculated as the interception of a 4-parameter logistic curve fit of OD values obtained from the 4-fold titration of the samples, starting from a 1:20 dilution to 1:81,920, with a cut off. The cut off was defined as
$$Cutoff=\bar{X}+SD\times f$$
where $\bar{X}$ is the mean of n blank determinations (*n* = 6), *SD* is the standard deviation, and *f* is the standard deviation multiplier provided by Frey et al. [36]. The confidence interval was set to 99.0%. The final titers were based on triplicate determinations. The curve fitting was carried out using GraphPad Prism and the cut off calculations and titer calculations were carried out using Excel.

## 3. Results

### 3.1. Interindividual Differences in Cross-Neutralization of SARS-CoV-2 Variants of Concern (VOCs)

To define the ability of individual DNA-vaccinated rabbit sera to cross-neutralize diverse SARS-CoV-2 variants, we first determined the neutralization titer for each of the five rabbits against the ancestral strain D614G as well as eight World Health Organization-defined SARS-CoV-2 VOCs (Figure 1) [33]. Based on the median neutralization titers calculated for all VOCs, the five rabbits could be categorized broadly into three groups: low (J044890: median VOC titer = 15), intermediate (J044904: median VOC titer = 33; J044906: median VOC titer = 51), and high (J044907: median VOC titer = 143; J044908: median VOC titer = 174) cross-neutralization. Despite having comparable anti-spike IgG levels (Appendix A), the five animals had distinct cross-neutralization profiles that would suggest a role for inter-individual-specific and/or dominant epitopes targeted within the polyclonal serum. To address this, a representative animal from each category was selected for an in-depth analysis of the B-cell epitopes targeted in the polyclonal sera, i.e., J044890, J044906, and J044908 (hereafter referred to as Rabbits 1, 2, and 3).

### 3.2. Polyclonal Antibody Binding Profiles of Serum with Variable Cross-Neutralization of the SARS-CoV-2 VOCs

We next characterized the antibody binding profiles of individual rabbit sera using a custom-made peptide microarray, presenting the cyclic constrained peptides spanning the complete index strain spike protein. The cyclic peptides may, to some extent, represent conformational epitopes better than linear peptides and were therefore selected for the analysis. To adjust for a potential confounding effect of inter-individual anti-spike IgG levels, an equivalent amount of total anti-spike IgG was tested for each rabbit, as determined from the ELISA binding IgG titers. We evaluated two different serum dilutions on the peptide microarray, at a 1:20 dilution to represent the peptide binding of low-affinity and/or low-abundance antibodies, and a 1:100 dilution to represent high-affinity and/or high-abundance antibodies.

As expected for the inter-individual polyclonal antibody responses, the animals had different serum antibody binding profiles (Figure 2A). At a serum dilution of 1:20, Rabbits 1, 2, and 3 targeted 210, 102, and 62 individual peptides, respectively. At a 5-fold higher serum dilution of 1:100, the three rabbits had fewer positive peptide detections at 77, 11, and 8, respectively. For both serum dilutions, Rabbit 2 and Rabbit 3, which had an intermediate and high cross-neutralization against the VOCs, targeted significantly fewer peptides and had distinct peptides in common (Figure 2B).

### 3.3. Dominant Targeting of an SD1 Epitope in Animals with Intermediate and High Cross-Neutralization of VOCs

The higher serum dilution of 1:100, which likely represents a detection of more abundant and/or higher affinity antibodies, identified two regions common between the two rabbits with VOC cross-neutralization, but not Rabbit 1, which had a low-to-undetectable neutralization against neutralization-resistant VOCs. These include a peptide in the SD1 within the S1 subunit, spanning amino acids 572–581 (TTDAVRDPQT), and a peptide in the connecting region of the S2 subunit, spanning amino acids 862–871 (PPLLTDEMIA) (Figure 3A). For Rabbit 3, the antibody binding to the peptide spanning residues 572–581 (TTDAVRDPQT) peaked above the assay upper limit of quantification at the 1:20 serum dilution (Figure 3B). At a 5-fold lower serum input on the microarray (1:100), the fluorescence intensity was near the upper limit of quantification (97.7% saturated), thus enabling a quantitative intra- and inter-individual comparison. Rabbit 2 similarly targeted the 572–581 peptide, but with a markedly lower intensity (27-fold; relative fluorescence units (RFUs) 3286 vs. 88,324), while Rabbit 1 did not target peptide 572–581 (RFU = 14), but rather an adjacent sequence spanning three peptides and residues 563–577 (QFGRDIADTTDAVR; RFUs = 922–2146). The level of antibody binding to peptide 572–581 correlated with the level of cross-neutralization of the VOCs (Figure 3C).

The other spike ectodomain peptide common between Rabbit 3 and Rabbit 2 includes the peptide spanning residues 862–871 (PPLLTDEMIA; RFUs at serum 1:100 = 911 and 255, respectively) (Figure 3D). The binding to this peptide was 97-fold (Rabbit 3) and 13-fold (Rabbit 2) lower compared to the binding measured for the aforementioned 572–581 peptide. Rabbit 1 had a low-level binding to peptide 862–871 (RFUs = 42). As observed for peptide 572–581, the level of antibody binding correlated with the level of the cross-neutralization of the VOCs (Figure 3E).

In summary, Rabbit 3, with the highest cross-neutralization of VOCs, shares common epitopes with Rabbit 2 that has a detectable, yet lower, cross-neutralization of neutralization-resistant VOCs. The level of antibody binding to both peptides (TTDAVRDPQT and PPLLTDEMIA) correlate with the level of cross-neutralization. However, proportionally, the magnitude of antibody binding to the peptide spanning residues 572–581 (TTDAVRDPQT) dominated the responses for both Rabbit 3 and Rabbit 2 (Figure 4). This SD1 epitope is highly conserved between all SARS-CoV-2 variants (Figure 5). In contrast to this targeted response, Rabbit 1 demonstrated a more diffuse antibody binding profile across the entire spike ectodomain. Beyond the shared epitopes between Rabbit 3 and Rabbit 2, Rabbit 3 furthermore had binding to a peptide in the RBD (residues 458–467; KSNLKPFERD; RFUs at serum 1:100 = 491) and immediately upstream of the S1/S2 furin cleavage site (residues 672–681; ASYQTQTNSP; RFUs = 841).

### 3.4. Epitopes Targeted within the Spike RBD

The RBD is a frequent target of neutralizing antibodies, including broadly cross-neutralizing antibodies such as S309 and CR3022. Anti-RBD antibodies are grouped into different classes (class 1–4) based on their blocking of the direct binding of ACE2 (class 1 and class 2), non-ACE2 blocking (class 3 and class 4), and binding to the specific conformation of RBD, i.e., up (class 1–4) or down (class 2 and 3) [37]. At a serum dilution of 1:20, we observed 32, 20, and 14 positive peptides within the RBD for Rabbits 1, 2 and 3, respectively. For the targets within the receptor binding motif (class 1 and class 2 antibodies), there were 15, 12, and 13 positive peptides with maximum RFUs of 2134, 1078 and 3809 for Rabbits 1, 2 and 3, respectively. For the targets of the class 3 and 4 antibodies, at a serum dilution of 1:20, there were 17, 8, and 1 positive peptides that included a maximum RFU of 854 observed in Rabbit 1 spanning amino acids 358–367 (ISNCVADYSV), 744 observed in Rabbit 2 spanning amino acids 508–517 (YRVVVLSFEL), and 747 observed in Rabbit 3 spanning amino acids 334–343 (NLCPFGEVFN). At a serum dilution of 1:100, we found 9, 1, and 1 positive peptides within the RBD for Rabbits 1, 2 and 3, respectively. For the targets within the receptor binding motif (class 1 and class 2 antibodies), there were 6, 1, and 1 positive peptides with maximum RFUs of 219, 111, and 491 for Rabbits 1, 2 and 3, respectively. For the targets of the class 3 and 4 antibodies, at a serum dilution of 1:100, there were three positive peptides in Rabbit 1 that included a maximum RFU of 257 spanning amino acids 358–367 (ISNCVADYSV).

### 3.5. The Effect of Beta VOC-Specific Spike Mutations on Peptide Binding

Each subarray included peptides with specific amino acid changes within the Beta spike protein. These include amino acid substitutions D80A, D215G, K417N, E484K, N501Y, D614G, and A701V, and deletion LAL241–243del. Of these, mutations K417N and E484K are known to induce neutralization resistance [38,39,40,41,42]. In comparison to the dominant antibody binding sites in the spike ectodomain, the antibody binding level to peptides spanning the aforementioned residues ranged from undetectable to low for both the index strain and Beta VOC (Figure 6). In all three animals, there was a low-level antibody binding to the index strain peptides spanning amino acid 484, which was reduced notably by the mutation in the Beta VOC. A very low binding to the K417N mutation was detected, with a modestly increased binding to the mutated peptides in all three animals. The antibody binding to D80A, LAL241–243del, N501Y, and A701V remained mostly unaffected by the amino acid changes. The analysis of Omicron-specific amino acid changes is precluded, since the microarray was designed and produced prior to the emergence of Omicron.

### 3.6. Binding Antibody Levels against the Spike Ectodomain and RBD

Since higher levels of antibody titers can lead to a cross-neutralization of resistant SARS-CoV-2 variants, we determined the levels of the anti-spike ectodomain IgG and the anti-RBD IgG relative to the levels of neutralizing antibodies against the D614G strain and the Beta VOC (Figure 7). Intriguingly, Rabbit 3, with the highest cross-neutralization, had the lowest spike ectodomain-binding IgG titers against the D614G spike (endpoint titer = 4584 vs. 7776 and 6294), and an even lower level of binding IgG titers against the Beta VOC spike ectodomain, although comparable to the other rabbits (endpoint titer = 2535 vs. 2747 and 4322) (Figure 7B). Anti-RBD IgG titers followed a similar pattern with endpoint titers against the D614G strain of 5087, 4377, and 3442 for Rabbits 1, 2, and 3, respectively (Figure 7C). Notably, Rabbit 3 together with Rabbit 1 had the lowest anti-RBD IgG binding titers for the Beta VOC (endpoint titers = 774 and 786, respectively, vs. Rabbit 2 = 2779) (Figure 7C).

A different ELISA format—the antigen–antibody–antigen sandwich ELISA—that detects the total antibodies of all isotypes against the index strain RBD, similarly showed that Rabbit 3 had the lowest RBD-specific total antibody levels (Figure 7D). To remove the possible confounding effect of high antibody titers on cross-neutralization, we normalized the virus neutralization titers using the binding IgG titers, which revealed that Rabbit 3 had the highest cross-neutralization capacity per unit of IgG compared to the other two rabbits (Figure 7E). In particular, compared to Rabbit 1, Rabbit 3 showed a 2.0-fold higher normalized neutralization capacity for the D614G strain and a 6.1-fold higher ratio for the Beta VOC.

## 4. Discussion

This study defines the serum antibody profiles of SARS-CoV-2 DNA-vaccinated rabbits with divergent cross-neutralization of SARS-CoV-2 VOCs using a peptide microarray displaying cyclic constrained peptides spanning the SARS-CoV-2 spike protein. The findings demonstrate that the cross-neutralization of variants is not necessarily the function of higher total-antibody responses, since the rabbit with the greatest cross-neutralization had the lowest anti-spike IgG binding antibody levels against the spike ectodomain and RBD. Furthermore, cross-neutralization did not appear to associate with an increased tolerance of amino acid changes known to induce neutralization resistance. Rather, we observed an IgG antibody response dominated by targeting an SD1 epitope in both animals with a moderate-to-high neutralization of VOCs, but not in an animal that lacked this functional antibody property.

On the spike S1 subunit, SD1 forms part of the hinge region and facilitates the RBD upward rotation, a conformational change required for the binding of the RBD to the ACE2 host receptor [43]. Despite the location outside the RBD, SD1 is a known target of neutralizing SARS-CoV-2 antibodies induced by infection and vaccination [16,17,18,22,23]. Evidence suggests that the mechanism of anti-SD1 antibody neutralization likely occurs through the conformational locking of the RBD in the ‘down’ conformation, and destabilizing spike trimers, rather than inducing S1 shedding or the stearic hindrance of ACE2 engagement [16,23]. In the present study, rabbit sera with a cross-neutralization of neutralization-resistant VOCs predominantly targeted an SD1 epitope (TTDAVRDPQT), which overlaps with that of two broadly neutralizing monoclonal antibodies [23]. The breadth of neutralization associated with anti-SD1 antibodies is attributable to this region of the spike protein being highly conserved between SARS-CoV-2 variants (Appendix A). Residues within this epitope (572–581) comprise one of the most conserved regions within the S1 subunit. The very low conservation scores indicate a highly conserved area among SARS and MERS-related viruses containing multiple residues that are believed to be well-suited for drug intervention [44,45]. It is therefore plausible that the prominent targeting of an SD1 epitope by the serum antibodies may have contributed to the observed cross-neutralization in Rabbit 3 and Rabbit 2. To validate the role of SD1 peptide 572–581 as a broadly neutralizing epitope, an appropriate approach would be to remove the antibodies targeting this site. In theory, this is achievable through antibody competition or antibody depletion. Antibody competition with a peptide, in this case a 10-mer peptide, does, however, pose a challenge, as the binding affinity is generally too low to compete with the binding to the full immunogen. Antibody depletion, using the circular 10-mer peptide to mimic the binding conditions present on the microarray, similarly pose challenges that could distort the interpretation. Slight changes in the epitope presentation, binding accessibility, linker chemistry, etc., could cause a drift in the targeted antibodies, resulting in an uncontrolled depletion and an incorrect evaluation of the effect on the neutralization capacity. In summary, our data show a correlation between SD1 antibody targeting and neutralization breadth. However, a validation of the contribution of these antibodies to the cross-neutralization capacity, supporting the findings in this study, is warranted.

Neutralizing anti-SD1 antibody epitopes may be structurally occluded in the prefusion state of the spike protein [16]. This brings into question the efficiency by which SARS-CoV-2 vaccines using the prefusion-stabilized spike protein, e.g., mRNA vaccines, induce neutralizing antibodies targeting this region. Of note, the majority of anti-SD1 neutralizing monoclonal antibodies were isolated from convalescent donors or those who experienced a breakthrough infection after mRNA vaccination, as well as a donor vaccinated with the native spike-encoding AZD1222 vaccine [16,18,22,23]. It is therefore of consequence that the SARS-CoV-2 pNTC-Spike DNA vaccine construct that induced the anti-SD1 antibodies in the rabbits also encodes a native spike protein, rather than a prefusion-stabilized protein [33]. It could be argued that expressing the native spike in vivo using a DNA vaccine is more representative of a natural virus infection compared to an mRNA vaccine using a prefusion-stabilized spike protein. It should be noted, though, that pre-fusion stabilization of the glycoprotein is regarded as being immunogenicity-enhancing, presumably because the stabilized glycoprotein cannot convert to the post-fusion structure or misfold, thereby preserving the immunogenic conformation [46,47,48].

We cannot exclude a possible contribution of antibodies targeting other spike regions sensitive to neutralization, such as the RBD, N-terminal domain, and fusion peptide. On the peptide microarray, the binding to peptides spanning the RBD was generally low, with the exception of residues 458–467 (KSNLKPFERD). The monoclonal antibody C98C7 that binds these residues has previously been reported to broadly neutralize VOCs including Omicron BA.1 [49]. RBD class 3 and 4 monoclonal antibodies C309 and CR3022 that bind epitopes immediately upstream of the receptor binding motif have similarly been shown to broadly neutralize VOCs including Omicron BA.1, BA.1.1, and BA.2 [50,51,52,53]. We found only a weak binding to this region, with a maximum RFU of 854 in all three rabbits. The general low-level binding to the entire RBD detected in this study may be attributable to the conformational structure of RBD that may be poorly represented on the peptide microarray [3], or to the effective glycan shielding limiting the antibody targeting of sites on the RBD [54]. Nonetheless, the low level of the anti-RBD IgG specific for the Beta VOC, and the total antibody levels to the D614G strain observed in Rabbit 3—the animal with the greatest level of cross-neutralization—would suggest that the antibody target that is responsible for the broad neutralization resides outside the RBD. The serum antibodies of the rabbit lacking cross-neutralization (Rabbit 1) targeted regions in the upstream helix and sub-domain 3, close to the heptad repeat 2. These two regions have previously been associated with neutralization breadth [51], although in Rabbit 1, specific peptides targeted in this region did not associate with the cross-neutralization of VOCs.

Taken together, this study demonstrates the combination of functional antibody data with epitope mapping using peptide microarray technology as a powerful approach to unveiling key antigenic sites with and without functional significance. A cyclic constrained peptide array, instead of a linear peptide array, may better mimic native protein folding, improving the comparison of array data with ELISA and live-virus neutralization assay data [55]. Discontinuous epitopes brought together from distant amino acid positions are, however, difficult to recapitulate with individual peptides, introducing a limitation of this approach. Nonetheless, with a small but well-selected group of animal subjects, it was possible in the polyclonal serum responses to identify a known, broadly neutralizing antibody target validated by studies of monoclonal antibodies targeting the same site. Those studies showed that this epitope reported to bind broadly cross-reactive monoclonal antibody sd1.040 [56,57], and 12–16 and 12–19 [23] are in fact discontinuous, giving indications that high-resolution cyclic peptide microarrays can identify antibodies targeting discontinuous conformational epitopes.

In our initial selection of the study population, we saw a relatively clear distinction in the cross-neutralization capacities and median of the VOC neutralization titers for each rabbit. Among the five rabbits tested, the two animals that were excluded from the epitope mapping, one with intermediate cross-neutralization and one with high cross-neutralization, had a high resemblance to the two animals included in each segment. We do, however, anticipate that there would be heterogeneity in the profiles of the antibody responses elicited from the remaining two rabbits, as we can see within the group of three rabbits. Including additional animals would highly likely show variations in antigenic targeting as well. Though we performed antibody profiling of only three animals, with each animal representing different categories of cross-neutralization efficiency, this study population displayed highly diverse antibody repertoires, unveiling key features that might be valuable to support future vaccine strategies.

This study contributes to the growing interest within functional and applied antibody research that supports findings on recently discovered SARS-CoV-2 broadly neutralizing epitopes. Combining the application of a DNA vaccine, encoding the full native spike protein with a high-resolution conformational peptide microarray spanning the full index strain spike protein adds a valuable novel perspective to the field, and provides insight into natural infection immunology. The identification of potential immunogenic epitopes that associate with broadly neutralizing antibodies may aid rational vaccine development, such as with the design of effective peptide-based vaccines. Vaccines containing multiple peptides triggering both a B-cell and a T-cell response, or DNA/RNA vectors containing synthetic genes encoding immunogenic epitopes, have promising perspectives and can be produced at a relatively low cost [58]. Peptide-based vaccines are, however, known to be less immunogenic, and the inclusion of an adjuvant to stimulate the immune response is therefore essential [58,59]. Peptide-based SARS-CoV-2 vaccines using the adjuvant lipopeptide synthetic TLR1/2 ligand XS15 [60] or aluminum phosphate (Adju-Phos^®^) [61] have been reported to induce potent cellular responses in neutralizing antibodies. Future analysis using this approach, including additional animals and other validation methods, e.g., vaccination challenge studies with the SD1 epitope, will further improve our understanding of these initial findings.

## Figures and Tables

**Figure 1 vaccines-11-01451-f001:**
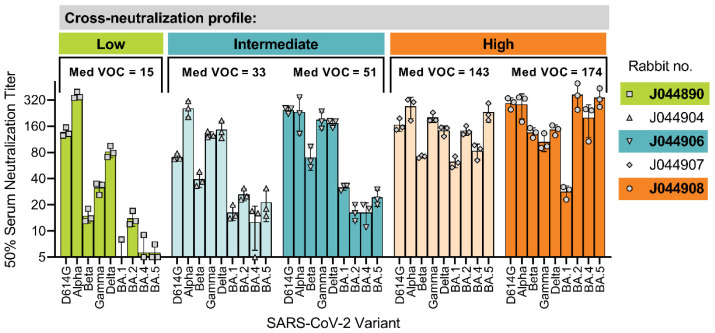
Cross-neutralization of serum antibodies from DNA-vaccinated rabbits against SARS-CoV-2 variants of concern. Rabbits received a DNA vaccine encoding the SARS-CoV-2 index strain spike protein three times with 2-week intervals [33]. Two weeks after final vaccination, 50% virus neutralization titers were determined for SARS-CoV-2 VOCs. Bars represent the mean virus neutralization titer for triplicate measurements against each variant and lines indicate standard deviation. Animals were categorized into low, intermediate, or high levels of cross-neutralization depending on their median neutralization titers against all VOCs. Med VOC was calculated as the median of all VOC neutralization titers for each rabbit. Animals selected for further analysis were J044890, J044906, and J044908 (hereafter referred to as Rabbits 1, 2, and 3).

**Figure 2 vaccines-11-01451-f002:**
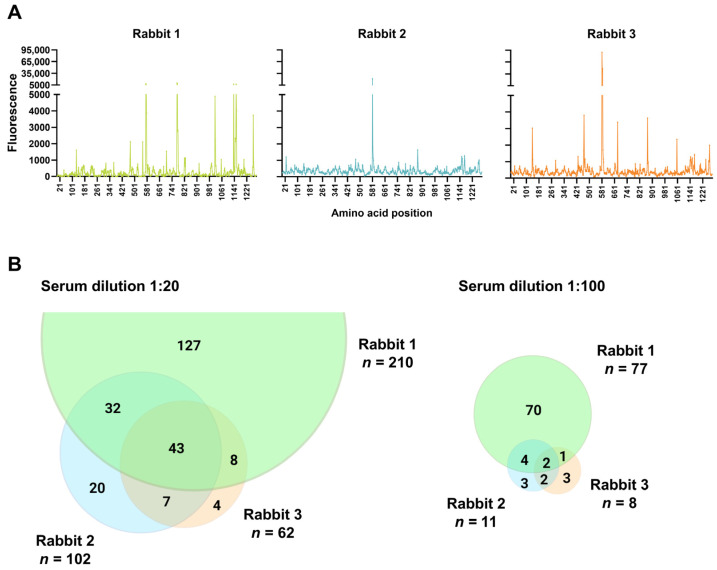
Serum antibody binding profiles targeting the SARS-CoV-2 spike protein. (**A**) Antibody binding of 1:20 diluted serum samples to 10-mer overlapping peptides spanning the ectodomain of the SARS-CoV-2 index strain spike protein. The *x*-axis represents the spike protein amino acid, as in the last position of the individual peptides. Fluorescence levels on the *y*-axis indicate the average level of antibody binding to each peptide measured in duplicate and are normalized according to a positive control included on each subarray. (**B**) The number of targeted peptides with fluorescent signals above a 99.9% confidence level cut-off value calculated from a negative control on each subarray. The Venn diagrams show the number of targeted peptides shared between the different animals and number specific to each animal.

**Figure 3 vaccines-11-01451-f003:**
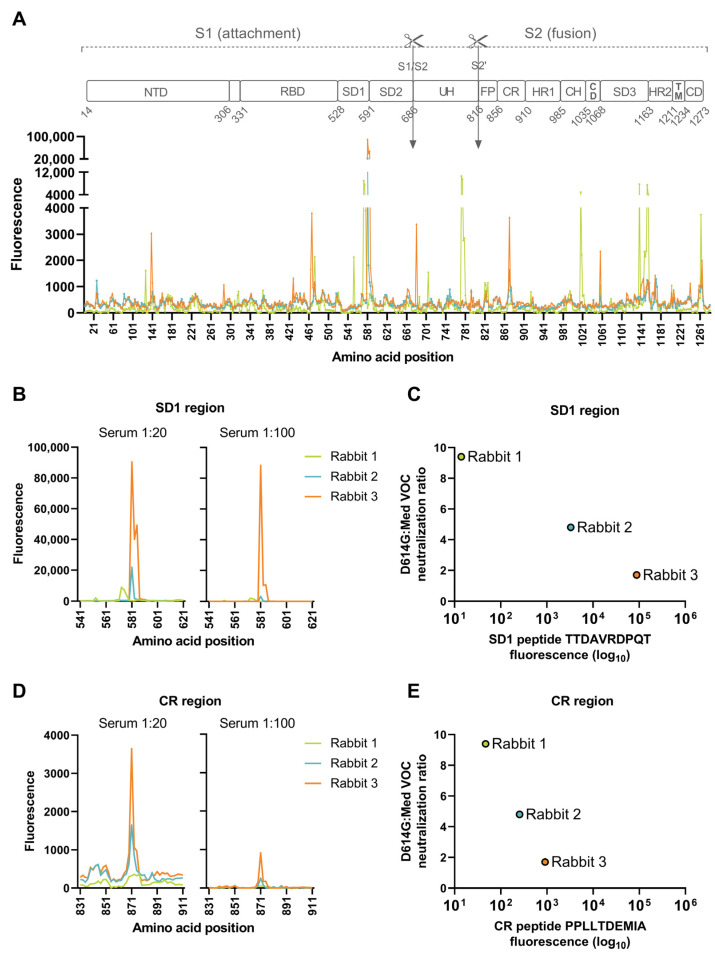
Dominant antibody binding to an SD1 epitope in animals with cross-neutralization of the VOCs. (**A**) Comparative serum antibody binding for the three animals relative to the different SARS-CoV-2 spike protein domains. S1 subunit (residues 14–685); S2 subunit (residues 686–1273); N-terminal domain (NTD); receptor binding domain (RBD); sub-domain 1, 2, 3 (SD1, SD2, SD3), S1/S2 furin cleavage site; upstream helix (UH); S2’ cleavage site; fusion peptide (FP); connecting region (CR); heptad repeat sequence 1 and 2 (HR1, HR2); central helix (CH); connector domain (CD); transmembrane helix (TM); and connector domain (CD). Antibody binding with a sample dilution of 1:20 and 1:100 to the SD1 region (**B**) and correlation analysis of antibody binding to the peptide TTDAVRDPQT (amino acids 572–581 on the index strain) to cross-neutralization capacity of the VOCs (**C**). Antibody binding with a sample dilution of 1:20 and 1:100 to the CR region (**D**) and correlation analysis of antibody binding to the peptide PPLLTDEMIA (amino acids 862–871 on the index strain) to cross-neutralization capacity of the VOCs (**E**). Antibody binding is expressed as fluorescence intensity measured for the duplicate spots of the peptide. Cross-neutralization capacity is calculated as the ratio of neutralization titers determined for the D614G strain to the median of neutralization titers of the VOCs (Med VOC).

**Figure 4 vaccines-11-01451-f004:**
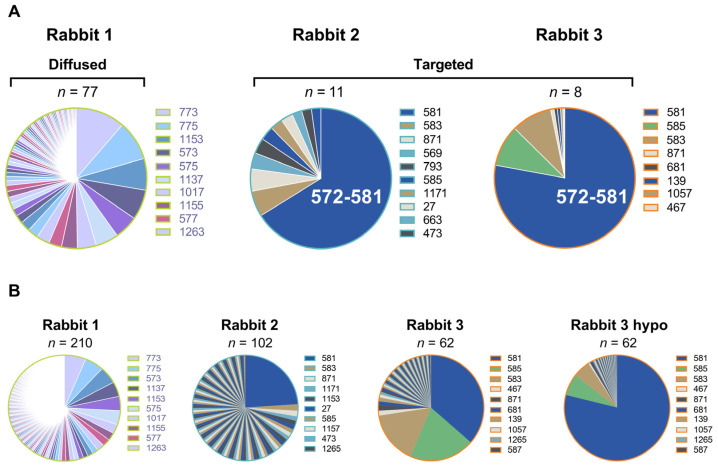
Proportionate binding of serum IgG to overlapping peptides spanning the SARS-CoV-2 index strain spike ectodomain for each rabbit. The pie charts consider the level of antibody binding, expressed as relative fluorescent units, to each peptide as a proportion of the sum of relative fluorescent units measured for binding to all overlapping peptides. (**A**) At a serum dilution of 1:100, Rabbit 2 and Rabbit 3 show dominant, targeted binding to the peptide spanning residues 572–581 (TTDAVRDPQT), while Rabbit 1 lacks binding to this peptide and show a more diffuse antibody binding profile. (**B**) At a serum dilution of 1:20, the dominance of peptide 572–581 remains for Rabbit 2 and Rabbit 3. Rabbit 3 hypo is the hypothetical proportion with no limiting spot saturation.

**Figure 5 vaccines-11-01451-f005:**
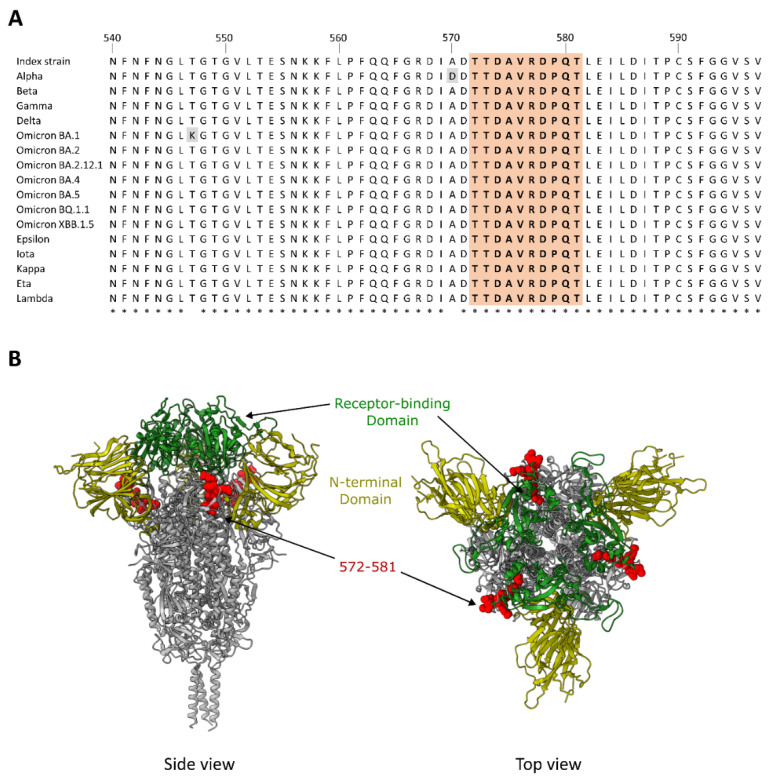
Sequence conservation of peptide 572–581 (TDAVRDPQT) and the position of the epitope in the SD1 regions of the full index strain spike protein trimer. (**A**) An amino acid sequence alignment of the partial SARS-CoV-2 spike protein encompassing the SD1 and peptide (highlighted in peach) targeted in animals with cross-neutralization capacity. (**B**) The position of the SD1 peptide on the crystal structure of the spike protein indicated in red spheres. The receptor binding domain is in green and the N-terminal domain is in beige. Asterisk (*) refers to identical amino acids in all sequences. Substitutions are indicated in grey.

**Figure 6 vaccines-11-01451-f006:**
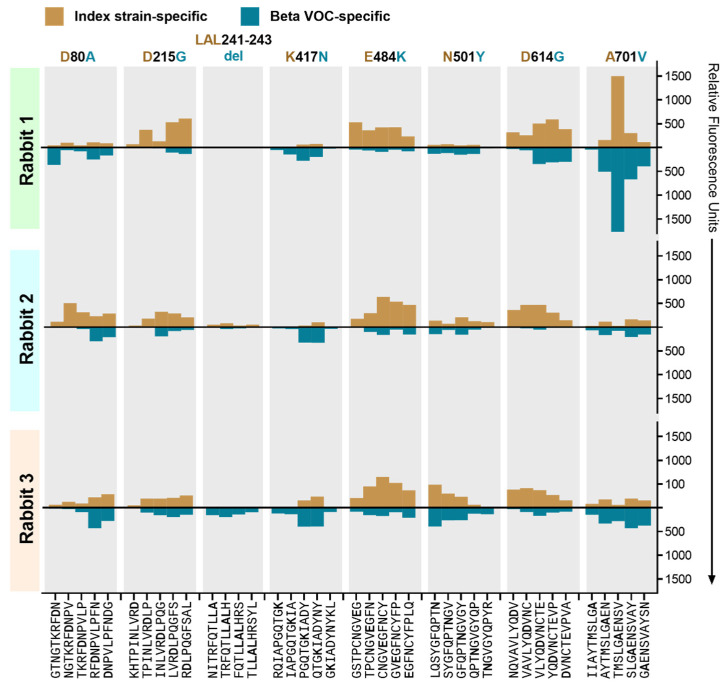
Antibody binding to overlapping peptides bearing SARS-CoV-2 Beta VOC spike protein variations relative to the index strain. Mirrored serum antibody binding to index strain peptide and their counterpart containing Beta-specific amino acid substitutions D80A, D215G, K417N, E484K, N501Y, D614G, and A701V, and deletion LAL241–243del. The *x*-axis indicates the 10-mer peptide sequence with the indicated amino acid variation at different positions; the third peptide for each variant bears the residue of interest approximately in the center of the sequence. The *y*-axis presents the mean fluorescence intensity minus background for duplicate peptide spots on the microarray. Values in gold indicate antibody binding to the index strain sequence and values in turquoise indicate antibody binding to the Beta VOC sequence.

**Figure 7 vaccines-11-01451-f007:**
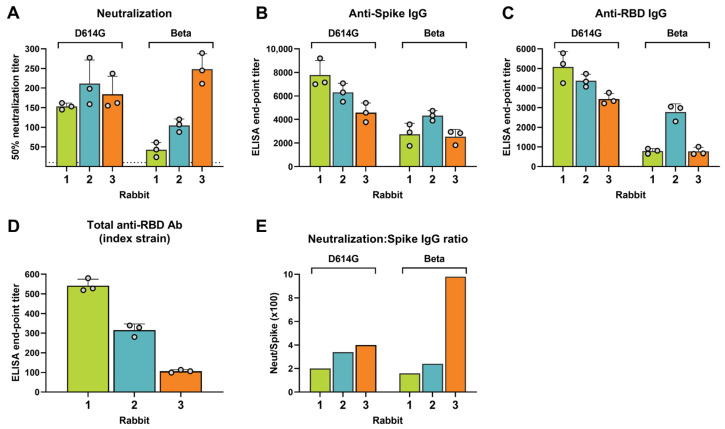
Virus neutralization and antibody binding of the SARS-CoV-2 D614G strain and Beta VOC. (**A**) The 50% live virus neutralization titers against the D614G strain and Beta VOC. Dotted line represents the lowest serum dilution tested. (**B**) Endpoint titers of binding IgG specific for the spike ectodomain of the D614G strain and Beta VOC, as measured in an indirect ELISA. (**C**) Endpoint titers of binding IgG specific for the spike protein receptor binding domain (RBD) of the D614G strain and the Beta VOC, as measured in an indirect ELISA. (**D**) Endpoint titers of total antibodies of all isotypes binding to the RBD of the index strain, as measured in a double antigen sandwich ELISA. (**E**) Virus neutralization normalized for IgG-binding antibody levels calculated as the neutralization titers divided by the anti-spike IgG titers. All error bars are standard deviation error bars.

## Data Availability

The datasets presented in this study can be found in online repositories. The name of the repository and accession number can be found at ArrayExpress; E-MTAB-13040.

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
