# Peer review of "A Candidate DNA Vaccine Encoding the Native SARS-CoV-2 Spike Protein Induces Anti-Subdomain 1 Antibodies"

_vaccines, 2023, doi:10.3390/vaccines11091451_

Round 1
Reviewer 1 Report
Frische et al. performed epitope mapping of sera from rabbits previously immunized with DNA-based SARS-CoV2 spike vaccine in this study. While these animals showed a similar neutralization activity against the original strain of SARS-CoV2, their sera have different capacities to cross-neutralize other variants of concern (VOC). To understand the difference in cross-neutralization, they have mapped the epitopes recognized by these antisera using an overlapping peptide array. The authors identified domains, such as subdomain 1 (SD1) and connecting region (CR), that are recognized by antisera with a higher cross-neutralization activity. These regions tend to be conserved among the original SARS-CoV2 and VOCs. Furthermore, in animals that have less cross-neutralization activity, the antiserum recognizes a more diffused pattern of peptides whereas in animals that have a higher cross-neutralization activity, their antisera have a more targeted recognition focussing around the region between amino acid residues 572-581 of SD1.
The finding of this paper is relevant to vaccine development and provides mechanistic understanding. As the authors have discussed, there is a limitation to using peptide arrays to map epitopes of antisera. Confirmation-specific epitopes are not examined in this study. While the authors have successfully identified a region recognized by antisera with a higher cross-neutralization, this correlation has not been further validated as the cause. Accordingly, experiments aiming to address mapping the confirmation-specific antibodies and/or whether antibodies targeting SD1 are sufficient for cross-neutralization against VOCs will strengthen this paper.
Major concerns:
1) Competition ELISA to assay if the rabbit antisera can compete binding against confirmation-specific antibodies of SARS-CoV2 will map the confirmation-specific epitopes in the antisera.
2) Depletion of SD1-specific antibodies using peptide encoding amino acid residues 572-581 and testing its subsequent neutralization could address if the antibody target to these regions is essential for cross-neutralization.
Author Response
Responses to Reviewers Comments
Independent Review Report, Reviewer 1
The finding of this paper is relevant to vaccine development and provides mechanistic understanding. As the authors have discussed, there is a limitation to using peptide arrays to map epitopes of antisera. Confirmation-specific epitopes are not examined in this study. While the authors have successfully identified a region recognized by antisera with a higher cross-neutralization, this correlation has not been further validated as the cause. Accordingly, experiments aiming to address mapping the confirmation-specific antibodies and/or whether antibodies targeting SD1 are sufficient for cross-neutralization against VOCs will strengthen this paper.
Response: We would like to thank the Reviewer for their time and constructive comments. We amended and improved the manuscript accordingly.
Major concerns:
- Competition ELISA to assay if the rabbit antisera can compete binding against confirmation-specific antibodies of SARS-CoV2 will map the confirmation-specific epitopes in the antisera.
- Depletion of SD1-specific antibodies using peptide encoding amino acid residues 572-581 and testing its subsequent neutralization could address if the antibody target to these regions is essential for cross-neutralization.
Response: We agree with the reviewer that an appropriate approach to assess if the identified SD1 epitope does in fact associate with neutralization breadth could be to evaluate the degree of competition from the peptide in a competition ELISA or to deplete antibodies targeting this epitope. We have also considered this.
However, experience with competition assays in our laboratory is, that small peptides, in this case a 10-mer circular peptide, does not match binding affinity from the actual immunogen. Since SD1 epitopes are discontinuous and the binding we detect likely only represent partial binding to the full epitope, we would expect to see very low competition from the peptide when in the presence of the full conformational, discontinuous epitope on the virus surface. In other words, due to weaker binding, the peptide will be displaced in the presence of the complete epitope on the spike protein. This will complicate evaluation of the contribution of P581 binding antibodies. For these reasons, we have not initiated a peptide-based competition laboratory investigation.
We equally agree with the reviewer, that depleting P581 antibodies could strengthen the conclusions drawn in this study. Unfortunately, we do not have a positive control peptide to indicate a successful depletion. This complicates evaluation of a depletion assay and would introduce uncertainties that could distort the final assessment of the effect on cross-neutralization. To create solid data using a depletion assay, to substantiate our hypothesis that neutralization breadth is proportional to the amount of SD1 binding antibodies, one must take multiple sources of variation into account such as comparable serum dilutions to control for confounding effect, bias from suboptimal peptide/antibody binding, epitope drift resulting from discrepancies in peptide presentation between microarray and depletion assay, differences in linker chemistry etc. For these reasons, briefly mentioned in the Discussion, we cannot present depletion study data.
We do however believe that several of the results presented in this manuscript point towards SD1 binding antibodies as having a central role in the observed neutralization breadth in our study population, e.g. 1. An observed correlation between levels of SD1 binding antibodies and neutralization breadth; 2. The highly conserved nature of this epitope; 3. Overlaps with two well-known monoclonal antibody broadly neutralizing epitopes; 4. A highly SD1 dominant response particularly observed in rabbit 3, animal with the superior cross-neutralization capacity.
Furthermore, we believe that our approach, where we by the use of microarray epitope mapping can detect significant levels of antibodies targeting a discontinuous epitope, does have promising perspectives for future vaccine research.
Revised manuscript, Page 14, Line 453 to 465: ‘It is therefore plausible that the prominent targeting of an SD1 epitope by serum antibodies contributed to the observed cross-neutralization in Rabbit 3 and Rabbit 2. To validate the role of SD1 peptide 572-581 as a broadly neutralizing epitope, an appropriate approach would be to remove antibodies targeting this site. In theory, this is achievable through antibody competition or antibody depletion. Antibody competition with a peptide, in this case a 10-mer peptide, does however pose a challenge as the binding affinity is generally too low to compete with binding to the full immunogen. Antibody depletion, using the circular 10-mer peptide to mimic binding conditions present on the microarray, similarly pose challenges that could distort the interpretation. Slight changes in epitope presentation, binding accessibility, linker chemistry etc. could cause a drift in the targeted antibodies resulting in uncontrolled depletion and an incorrect evaluation of the effect on neutralization capacity.’
Reviewer 2 Report
The manuscript by Frische is interesting and shows that rabbits produce specific antibodies after immunization with a candidate DNA vaccine.
Certainly, this article is publishabel, however, it needs some minor changes, as it follows:
1 - Detail the method of vaccine production and immunization. Please, add more details about control animals (injected with PBS, only plasmid)?
2 - Authors worked with 5 animals, and in certain experiments only 3. Data were described individually about the efficacy of neutralising antibodies. At least for me, a vaccine should be efficient in a larger group and be reproductive (after repetitions, i.e N=3). So, at this point I believe that the authors should highlight, for example in discussion, that the N is low and that 5 animals responded differently. Thus, there's a possibility that this vaccine is not as effective as expected.
Author Response
Responses to Reviewers Comments
Independent Review Report, Reviewer 2
The manuscript by Frische is interesting and shows that rabbits produce specific antibodies after immunization with a candidate DNA vaccine.
Certainly, this article is publishable, however, it needs some minor changes, as it follows:
Response: We would like to thank the Reviewer for their time and constructive comments. We amended and improved the manuscript accordingly.
1 - Detail the method of vaccine production and immunization. Please, add more details about control animals (injected with PBS, only plasmid)?
Response: We agree with the reviewer that the manuscript could benefit from a higher degree of detail describing the procedure of which the animals were vaccinated. This has been described in detail in the paper describing the development of the vaccine, Lassauniere et al (1), however we have also added more detail to this manuscript in the Materials and Methods section. We hope this addition provides the necessary details to fully evaluate the results provided.
Revised manuscript, Page 2-3, Line 89 to 100: ‘We developed and performed a preclinical evaluation of a candidate DNA vaccine in different animal models, including rabbits [33]. Five ten-week-old New Zealand white rabbits each received 125 µg of a nanoplasmid vector three times at two-week intervals. This DNA vaccine encodes the full, unmodified SARS-CoV-2 index-strain spike protein derived from the Wuhan-Hu-1 strain (MN908947). The human codon optimized SARS-CoV-2 spike sequence was synthesized by GeneArt (Thermo Fisher Scientific, Germany) and subcloned using EcoRI and XhoI into the NTC8685-eRNA41H vector backbone (Nature Technology Corporation, Lincoln, NE, USA). NTC8685-eRNA41H encodes a RIG-I agonist that stimulates a type 1 interferon response that improves antibody and cellular responses. The pNTC-Spike vector was produced by Nature Technology Corporation using an RNA-OUT antibiotic-free selection procedure in NTC4862 E. coli cells (DH5α attλ::P5/6 6/6-RNA-IN-SacV, Cmr) at 10 mg/mL in phosphate buffered saline (PBS).’
2 - Authors worked with 5 animals, and in certain experiments only 3. Data were described individually about the efficacy of neutralising antibodies. At least for me, a vaccine should be efficient in a larger group and be reproductive (after repetitions, i.e N=3). So, at this point I believe that the authors should highlight, for example in discussion, that the N is low and that 5 animals responded differently. Thus, there's a possibility that this vaccine is not as effective as expected.
Response: The 5 animals were used to evaluate the immunogenicity of the DNA vaccine, it was not a challenge study to determine protection/efficacy. Rabbits were selected due to their size and skin type for evaluation of the preferred vaccine delivery method – needle-free intramuscular administration. Due to the cost of rabbit experiments and our experience, we typically find that 5 animals are sufficient to show immunogenicity of a DNA vaccine. Furthermore, the vaccine-homologous immune responses e.g. D614G virus neutralization titers, were very similar between the animals, it is the cross-neutralization that differs due to inter-individual antibody profiles. We agree that the cross-neutralization will not be consistent in all subjects who receive the DNA vaccine. However, it is of consequence that a DNA vaccine encoding a wild-type spike protein can induce SD1 antibodies. It is therefore informative for rational vaccine design of future broadly reactive SARS-CoV-2 vaccines.
We addressed the reviewer’s concern in the discussion as follows:
Revised manuscript, Page 15-16, Lines 515 to 526: ‘In our initial selection of study population, we saw a relatively clear distinction in cross-neutralization capacities and median of VOC neutralization titers for each rabbit. Among the 5 rabbits tested, the two animals excluded from epitope mapping, one with intermediate cross-neutralization and one with high cross-neutralization, had high resemblance with the two animals included in each segment. We do however anticipate that there would be heterogeneity in the profiles of the antibody responses elicited from the remaining two rabbits as we see within the group of 3 rabbits. Including additional animals would highly likely show variations in antigenic targeting as well. Though we performed antibody profiling of only three animals, each animal representing different categories of cross-neutralization efficiency, this study population displayed highly diverse antibody repertoires unveiling key features that might be valuable to support future vaccine strategies.’

Round 2
Reviewer 1 Report
The revised version of the manuscript did not address the concern about the limitation of using peptide array to define B cell epitopes. Despite using the cyclic-constrained peptide, antibodies targeting confirmation-dependent epitopes or epitopes that require proper folding of the antigen are missed by this analysis. This limitation should be discussed and included in the manuscript.
While they have concluded that SD1-targeting antibodies could provide cross-neutralization protection, this finding is correlative at this point. Without further experimentation to directly address if SD-1 targeting antibodies are responsible for cross-neutralization, this has weakened the support of the main finding in this manuscript. In the revised version of the manuscript, the authors have included reasons why the suggested experiments might fail, but they have not offered an alternate way to address if SD-1 targeting antibodies is essential. Therefore, the finding of this manuscript remains correlative.
Author Response
Responses to Reviewers Comments
Independent Review Report, Reviewer 1
Response: We once again would like to thank the Reviewer for their time and constructive comments.
The revised version of the manuscript did not address the concern about the limitation of using peptide array to define B cell epitopes. Despite using the cyclic-constrained peptide, antibodies targeting confirmation-dependent epitopes or epitopes that require proper folding of the antigen are missed by this analysis. This limitation should be discussed and included in the manuscript.
Response: We appreciate that a peptide microarray cannot fully represent the conformational landscape of epitopes present on a protein. In light of this understanding, we believe that we have already addressed this particular limitation i.e. the peptide array is not completely representative of the full antibody binding profile, in different contexts in the Results section and in the Discussion as follows:
Page 6-7, line 262-264: ‘The cyclic peptides may, to some extent, represent conformational epitopes better than linear peptides and was therefore selected for the analysis.’
Page 15, line 495-499: 'The general low-level binding to the entire RBD detected in this study may be attributable to the conformational structure of RBD that may be poorly represented on the peptide microarray [3], or effective glycan shielding limiting antibody targeting of sites on the RBD [55].'
Page 15, line 512-514: 'Discontinuous epitopes brought together from distant amino acid positions are, however, difficult to recapitulate with individual peptides, introducing a limitation of this approach.'
While they have concluded that SD1-targeting antibodies could provide cross-neutralization protection, this finding is correlative at this point. Without further experimentation to directly address if SD-1 targeting antibodies are responsible for cross-neutralization, this has weakened the support of the main finding in this manuscript. In the revised version of the manuscript, the authors have included reasons why the suggested experiments might fail, but they have not offered an alternate way to address if SD-1 targeting antibodies is essential. Therefore, the finding of this manuscript remains correlative.
Response: We appreciate that, lacking a validation method, it cannot be conclusively stated that the SD1 binding is the only contributor to the observed cross-neutralization. We were therefore very careful with our word choice from the outset, stating clearly that it is an association (not causation) between the SD1 targeting and the observed broad neutralization. To further clarify that a definitive causative role of the SD1 antibodies, but rather an association, is not our conclusion, we have rephrased the following sentences, and added to the Discussion that validation is warranted.
Revised manuscript, Page 14, line 456-458: ‘It is therefore plausible that the prominent targeting of an SD1 epitope by serum antibodies may have contributed to the observed cross-neutralization in Rabbit 3 and Rabbit 2.’
Revised manuscript, Page 16, line 538-540: ‘The identification of potential immunogenic epitopes that associates with broadly neutralizing antibodies may aid rational vaccine development such as designing effective peptide-based vaccines.’
Revised manuscript (addition), Page 14, line 468-471: ‘In summary, our data show a correlation between SD1 antibody targeting and neutralization breadth. However, a validation of the contribution of these antibodies to the cross-neutralization capacity, supporting the findings in this study, is warranted.’

Round 3
Reviewer 1 Report
Thank you for address my comments. Good luck on your research.